# Prompt Learning with Optimal Transport for Vision-Language Models

## Abstract

With the increasing attention to large vision-language models such as CLIP, there has been a significant amount of effort dedicated to building efficient prompts. Unlike conventional methods of only learning one single prompt, we propose to learn multiple comprehensive prompts to describe diverse characteristics of categories such as intrinsic attributes or extrinsic contexts. However, directly matching each prompt to the same visual feature is problematic, as it pushes the prompts to converge to one point. To solve this problem, we propose to apply optimal transport to match the vision and text modalities. Specifically, we first model images and the categories with visual and textual feature sets. Then, we apply a two-stage optimization strategy to learn the prompts. In the inner loop, we optimize the optimal transport distance to align visual features and prompts by the Sinkhorn algorithm, while in the outer loop, we learn the prompts by this distance from the supervised data. Extensive experiments are conducted on the few-shot recognition task and the improvement demonstrates the superiority of our method.

## 1 Introduction

In the past few years, large-scale vision-language pre-trained (VLP) models, such as CLIP [39], ALIGN [17], and BLIP [23] have achieved remarkable success in open-world visual concept learning. These methods have brought new light but also pose a new question: how to efficiently adapt the knowledge from pretraining to the downstream tasks since these models are typical of massive sizes which are not feasible for normal users to re-train.

One of the conventional paradigms of utilizing pretrained knowledge is "pre-training, fine-tuning", which fixes the architecture of the pre-trained neural network and tunes its parameters using task-specific objective functions. Beyond fine-tuning the parameters, many recent methods [63, 64] introduce the concept of prompt

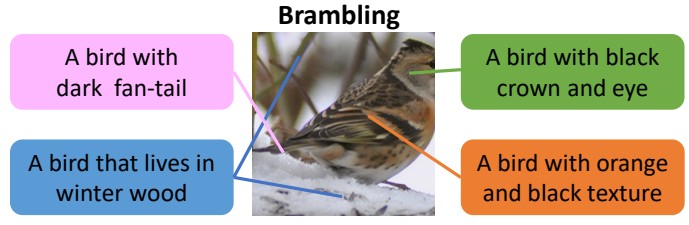

**Brambling**

A bird with dark fan-tail

A bird with black crown and eye

A bird that lives in winter wood

A bird with orange and black texture

Figure 1: The motivation that one category can be complementarily described in different views (An example of "Brambling").

learning from the field of NLP to the vision domain and achieve striking performance gain for the few-shot visual classification. They fix the model parameters and instead learn suitable prompts by turning a template sentence into a set of learnable vectors. Then, these prompts are learned by minimizing the distance between the visual features and prompt-based language features.

Despite significant improvements over manual prompts, learning only a sentence is intuitively insufficient to represent a class. One class can be described by many intrinsic characteristics and

even extrinsic context relations. Thus, for one object, we may have multiple prompt candidates which focus on different attributes. As shown in Figure 1, we can describe the class "Brambling" in different views: such as the color of the wing, the color of the crown and eyes, the shape and color of the tail, and even the living environment information. It motivates us to learn multiple prompts to comprehensively represent the class and thus facilitate classification.

The most natural solution is to directly learn multiple prompts by respectively matching each prompt with the visual features. However, it is the same as matching the mean of prompt features and the visual features. This solution is problematic since all prompts are encouraged to be closer to one single point and thus tend to learn the same characteristics. It contradicts our purpose to learn comprehensive prompts. To solve this problem, we tested adding some constraints to push away the prompt from each other, but found that this solution still fails to learn representative and comprehensive prompts. This solution treats the visual representation as one single point, and such a unified view of visual features ignores the fact that different prompts may only focus on one or a subset of characteristics.

To address this problem, in this paper, we propose Prompt Learning with Optimal Transport (PLOT), which applies optimal transport (OT) to align the local visual features and multiple textual prompts. Optimal transport can calculate the distance between two distributions under the form of multiple sampling. In our prompt learning framework, we formulate local visual features and multiple prompts as the samplings of two discrete distributions and use OT to encourage fine-grained cross-modal matching. Specifically, to obtain the local visual features with different semantic clues, we extract all feature maps as the visual representation instead of the single global representation. Fortunately, we can easily obtain the visual feature maps from the visual encoder of CLIP by using all outputs of the multi-head self-attention layer [42]. Then the problem comes down to how to calculate the distance between two feature sets.

We solve this problem by introducing the optimal transport theory [51] and formulate the feature sets as a discrete probability distribution where each feature has an equal probability value. Furthermore, to reduce the computational cost and avoid the extra model parameters, we learn the prompts with a two-stage optimization strategy. At the first stage in the inner loop, we fix both visual and text features and optimize the optimal transport problem by a fast Sinkhorn distances algorithm [6]. Then, in the outer loop, we fix all parameters of optimal transport and back-propagate the gradient to learn the prompts with different characteristics. *Compared with conventional distance (such as Euclidean distance of mean features), optimal transport can align different visual features for each local prompt, which is more robust to the visual misalignment and tolerates well feature shift [44]. It is because OT learns an adaptive transport plan to align features, which achieves fine-grained matching across two modalities.* We conduct experiments on 11 datasets following the standard setting of CLIP [39] and CoOp [63] to evaluate our method. These experiments span the visual classification of generic objects, scenes, actions, fine-grained categories, and so on. The significant result improvement demonstrates that PLOT can effectively learn representative and comprehensive prompts.

## 2 Related Work

**Optimal Transport** The Optimal Transport [30] is initially introduced to solve the problem of how to reduce the cost when moving several items simultaneously. Recently, OT theory has drawn wide attention in the machine learning and computer vision community by comparing distributions readily available to them under the form of feature sets [37]. Due to the brilliant property of distribution matching, OT has been applied in many theoretic and application tasks including generative models [1, 45, 60], structural matching [4, 57, 61, 56] (e.g. sequence matching [4] and graph matching [56]), and other distribution-based tasks (such as clustering [22], distribution estimation [2], and causal discovery [50]). *In this paper, we use OT to align the features of vision and language modalities which represents the data structure by learning an adaptive transport plan [44].*

**Vision-Language Pre-trained Models** Vision-Language Pre-trained (VLP) models aim to explore the semantic correspondence between the vision and language modalities through large-scale pre-training. Recently, VLP models have achieved an exciting performance improvement in the zero-shot and few-shot visual recognition [39, 10, 63, 64, 59], which shows the great potential to promote open-world visual understanding with the help of language. One key part of learning VLP models is the self-supervised learning objective on two modalities. The popular VLP objectives can be divided into reconstruction [25, 15, 8, 20], contrastive matching [39, 17, 16], or the combination of both two [24, 54, 19]. Besides, recent progress in the field of VLP also benefits a lot from large-scale

pair-wised datasets. For example, CLIP [39] applies 400 million image-text pairs for contrastive learning, while ALIGN even exploits 1.8 billion data pairs. Beyond recognition, these VLP models also show great potential for other downstream applications, such as dense prediction [42, 62], image generation [31, 41, 35], and action understanding [53, 48].

**Prompt Learning** Prompt learning is introduced from the field of NLP to efficiently adapt the large language model to downstream tasks. Different from the conventional "pre-training, fine-tuning" paradigm which initializes the pre-trained model and tunes the parameters of the network using downstream task-specific objective functions, prompt learning applies textual prompt to reformulate the downstream tasks as the original pretrained task [27, 36]. By the prompt, the domain shift between pretrained task and downstream application is reduced and thus the pretrained knowledge can be easier adapted to downstream tasks. The concept of prompt learning [36, 40, 38] begins from the success of GPT [40] series. Early prompt learning methods (such as Petroni *et al.* [36] and Pörner *et al.* [38]) always manually create templates based on human prior knowledge. Furthermore, some mining-based methods [18] and gradient-based methods [46] are proposed to automatically search for appropriate templates. Beyond search in the discrete space, some methods [26, 49, 28] remove the constraint that the prompts are "words" and instead learn prompts in the continuous embedding space. Recently, CoOp [63] and its extended version [64] introduce prompt learning into open-world visual understanding to adapt the knowledge from the large-scale visual-language pretrained models and achieve great performance improvement on the few-shot visual recognition. Compared with CoOp, our PLOT method further improves prompt learning by introducing the optimal transport distance to learn multiple local prompts and achieves fine-grained vision-language matching.

## 3 Approach

In this section we will first revisit the baseline method CoOp 3.1, review the preliminaries of optimal transport 3.2, and then introduce our proposed PLOT 3.3 to show how we can learn multiple comprehensive prompts.

### 3.1 A Revisit of CoOp

CoOp [63] is one of the pioneering methods to learn the prompts for using vision language pretrained knowledge (such as CLIP [39]) for downstream open-world visual recognition. Different from CLIP which manually designs the prompt templates, CoOp sets a part of context words in the template as continuous learnable parameters which can be learned from the few-shot data. Then the classification weights can be represented by the distance between the learned prompt and visual feature.

Specifically, given an image $x$, a visual feature $f = f(x)$ is obtained by the visual encoder $f$ of CLIP. Then, the textual prompt can be formulated as $t_k = \{vec_1, vec_2, \ldots, vec_L, c_k\}$, where $c_k$ is the word embedding of the class name, $\{vec_l|_{l=1}^L\}$ are learnable vectors with the same dimension as the original word embedding and L is the length of context words. With prompt $t_k$ as the input, the text encoder $g$ outputs the textual feature as $g_k = g(t_k)$. The final prediction probability is computed by the matching score as follows:

$$p(y = k|x) = \frac{exp(\text{sim}(f, g_k)/\tau)}{\sum_{k'=1}^K exp(\text{sim}(f, g_{k'})/\tau)},$$ (1)

where $\text{sim}(\cdot, \cdot)$ denotes a metric function such as cosine similarity, and $\tau$ stands for the temperature of Softmax. Then we can optimize the parameters of $\{vec_l|_{l=1}^L\}$ with the cross-entropy loss between the prediction and the labeled target.

### 3.2 Optimal Transport

Optimal transport (OT) distance is a widely used metric for the comparison of distributions. Here, we only focus on the discrete situation which is more related to our framework. Assuming we have two sets of points (features), the discrete distributions are formulated as:

$$U = \sum_{m=1}^M u_m \delta_{f_m} \qquad \text{and} \qquad V = \sum_{n=1}^N v_n \delta_{g_n},$$ (2)

where $u$ and $v$ are the discrete probability vectors that sum to 1, and $\delta_f$ is a Dirac delta function placed at support point $f$ in the embedding space. Then, the total distance of these two distributions

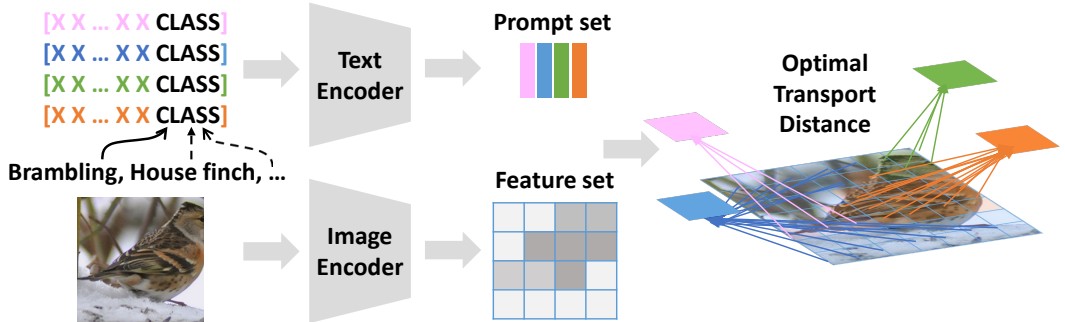

Figure 2: The framework of PLOT. PLOT first describes each category with multiple prompts and obtains a set of prompt features by text encoder. The image is also encoded as a set of local features. Then the optimal transport is used as the metric between prompts and visual features.

are written as:

$$< \boldsymbol{T}, \boldsymbol{C} > = \sum_{m=1}^{M} \sum_{n=1}^{N} \boldsymbol{T}_{m,n} \boldsymbol{C}_{m,n}. \tag{3}$$

We call $\boldsymbol{C}$ the cost matrix in which each point denotes the cost between $\boldsymbol{f}_m$ and $\boldsymbol{g}_n$, such as $\boldsymbol{C}_{m,n} = 1 - \sim(\boldsymbol{f}_m, \boldsymbol{g}_n)$. While the $\boldsymbol{T}$ is called the transport plan, which is learned to minimize the total distance. The optimization problem of optimal transport is formulated as:

$$d_{OT}(\boldsymbol{u}, \boldsymbol{v}|\boldsymbol{C}) = \underset{\boldsymbol{T}}{\text{minimize}} < \boldsymbol{T}, \boldsymbol{C} >$$
$$\text{subject to} \quad \boldsymbol{T}\mathbf{1} = \boldsymbol{u}, \boldsymbol{T}^T\mathbf{1} = \boldsymbol{v}, \boldsymbol{T} \geq 0. \tag{4}$$

As directly optimizing the above objective is always time-consuming, we apply the Sinkhorn distance [6] to use an entropic constraint for fast optimization. The optimization problem with a Lagrange multiplier of the entropy constraint is:

$$d_{OT,\lambda}(\boldsymbol{u}, \boldsymbol{v}|\boldsymbol{C}) = \underset{\boldsymbol{T}}{\text{minimize}} < \boldsymbol{T}, \boldsymbol{C} > -\lambda h(\boldsymbol{T})$$
$$\text{subject to} \quad \boldsymbol{T}\mathbf{1} = \boldsymbol{u}, \boldsymbol{T}^T\mathbf{1} = \boldsymbol{v}, \tag{5}$$

where $h(\cdot)$ is entropy and $\lambda \geq 0$ is a hyper-parameter. Then we can have a fast optimization solution with a few iterations as:

$$\boldsymbol{T}^* = \text{diag}(\boldsymbol{u}^t) exp(-\boldsymbol{C}/\lambda)\text{diag}(\boldsymbol{v}^t), \tag{6}$$

where $t$ denotes iteration and in each iteration $\boldsymbol{u}^t = \boldsymbol{u}/((exp(-\boldsymbol{C}/\lambda)\boldsymbol{v}^{t-1})$ and $\boldsymbol{v}^t = \boldsymbol{v}/((exp(-\boldsymbol{C}/\lambda)^T\boldsymbol{u}^t)$, with the initiation $\boldsymbol{v}^0 = \mathbf{1}$.

### 3.3 Prompt Learning with Optimal Transport

In this subsection, we introduce the details of our PLOT, which learns multiple prompts to describe different characteristics of the category by minimizing the OT distance.

Specifically, as shown in Figure 2, given an image $\boldsymbol{x}$, we first feed it to the visual encoder branch of CLIP. Apart from the global visual feature $\boldsymbol{f}$, we can also obtain a set of local features $\{\boldsymbol{f}_m|_{m=1}^M\}$. The visual encoder has a multi-head attention pooling layer in which the input is the combination of the global feature and a set of local features (feature map) and the output is a tensor with the shape $\mathbb{R}^{(H \times W+1) \times C}$, where $H$ and $W$ is the height and width of feature map and $C$ is the feature dimension. Therefore, we can obtain $M = H \times W$ local features and a global feature. At the same time, for class $k$, we can initialize N local prompts as $\{\boldsymbol{t}_{k,n}|_{n=1}^N\}$ with learnable vectors $\{\boldsymbol{vec}_{l,n}|_{l=1,n=1}^{L,N}\}$, where each is the same as the prompt in CoOp. With both visual and textual encoders, we can obtain local visual features $\boldsymbol{F} = \{\boldsymbol{f}_m|_{m=1}^M\} \in \mathbb{R}^{M \times C}$ and prompt features $\boldsymbol{G}_k = \{\boldsymbol{g}_n|_{n=1}^N\} \in \mathbb{R}^{N \times C}$.

In the inner loop, we learn the transport plan $\boldsymbol{T}$ with these fixed support sets $\boldsymbol{F}, \boldsymbol{G}_k$, by minimizing the following OT distance to push $\boldsymbol{G}_k$ to $\boldsymbol{F}$:

$$d_{OT}(k) = d_{OT}(\boldsymbol{u}, \boldsymbol{v}|\mathbf{1} - \boldsymbol{F}^T\boldsymbol{G}_k), \tag{7}$$

where $\boldsymbol{C} = \mathbf{1} - \boldsymbol{F}^T\boldsymbol{G}_k$ denotes that we use the cosine distance between $\boldsymbol{F}$ and $\boldsymbol{G}_k$ as the cost matrix. Then we can obtain the solution of transport plan $\boldsymbol{T}^*$ as Eq (6) and the final OT distance $d_OT(k)$.

Given the OT distance between $\boldsymbol{G}_k$ and $\boldsymbol{F}$, we reformulate the prediction probability as:

$$p_{ot}(y = k|\boldsymbol{x}) = \frac{exp((1 - d_{OT}(k))/\tau)}{\sum_{k'=1}^{K} exp((1 - d_{OT}(k'))/\tau)}. \tag{8}$$

In the outer loop, we fix the transport plan $\boldsymbol{T}^*$ and apply the cross entropy loss to optimize the $\{\boldsymbol{vec}_{l,n}|_{l=1,n=1}^{L,N}\}$ as:

$$L_{CE} = -\frac{1}{|\mathcal{X}|} \sum_{\boldsymbol{x} \in \mathcal{X}} \sum_{k=1}^{K} y_{\boldsymbol{x},k} p_{ot}(y = k|\boldsymbol{x}), \tag{9}$$

where $\boldsymbol{y_x}$ is a one-hot label vector. The detail algorithm can be found in the supplementary materials.

*Though the optimization strategy of the optimal transport and prompts is two-stage, the whole training flow is end-to-end. It is because that the transport plan is computed using a small number of matrix multiplications as one forward module of the neural network. The gradients of these matrix multiplications are taped for backpropagation for end-to-end optimization, which makes the whole system fully differentiable (including the iterative algorithm) and easy to implement using an autograd library like PyTorch. In the experiments, we found that it is natural and relatively easy to this optimization strategy.*

### 3.4 Inference strategy

*In the inference, given one query image and the learned prompts, we first obtain the a visual feature set containing $M = H \times W$ vectors and a prompt feature set containing $N \times C$ vectors. Then, we calculate the distance between the visual feature set and the prompt feature set of each class by OT as (6). After obtaining the OT distance for each class, we sort the distance and classify the image.*

## 4 Experiments

Extensive experiments are conducted to evaluate our method, including comparison with CoOp, ablation studies, parameter analysis extensibility analysis, computing cost analysis and visualization.

### 4.1 Datasets

We followed the experimental settings in the CoOp [63] for the few-shot learning evaluation. The experiments are conducted on the 11 visual recognition datasets, including Caltech101 [9], DTD [5], EuroSAT [12], FGVCAircraft [29], Flowers102 [32], Food101 [3], ImageNet [7], OxfordPets [33], StanfordCars [21], SUN397 [55], and UCF101 [47]. These datasets span visual classification of generic objects, scenes, actions, fine-grained categories, and so on, which constitutes a comprehensive evaluation of our method. All experiments adopted the few-shot evaluation protocol used in CLIP [39] and CoOp [63], where we respectively choose 1, 2, 4, 8, and 16 shots for model training and use the original test set for evaluation. Besides, we also evaluated the robustness of our method with domain shift. Following CoOp, we used the ImageNet as the source domain and evaluate our method with ImageNet-based robustness evaluation datasets including ImageNetV2 [43], ImageNet-Sketch [52], ImageNet-A [14], and ImageNet-R [13]. A detailed introduction of each dataset can be found in the supplementary materials.

### 4.2 Implementation details

We chose CoOp [63] as our main competitor to evaluate our method. Compared with CoOp which only learns a global prompt for one class, our PLOT method learns multiple local prompts and applies the OT distance for better fine-grained alignment. Besides, we also reported the performance of training a linear classifier with the CLIP [39] features. It is also a widely-used strategy to adapt the pretrained knowledge for the downstream task [46]. We reproduced the performance of CoOp and the CLIP linear probe with the released official code.

The original CoOp method has different versions with different class token positions and parameter initialization strategies. We applied the default model that fixes the class token positions in the end due to the limited performance gap between two different ways of positioning the class token. Besides, we used the random parameter initialization strategy but not the class-specific context version. Following the widely used setting in [63, 64, 10, 58], we also chose RN50 [11] as the backbone network of the visual branch and set the length of learnable context tokens as 16. All the code of our method is based on CoOp, which adopted the SGD optimizer with 0.002 initial learning rate, CosineAnnealingLR

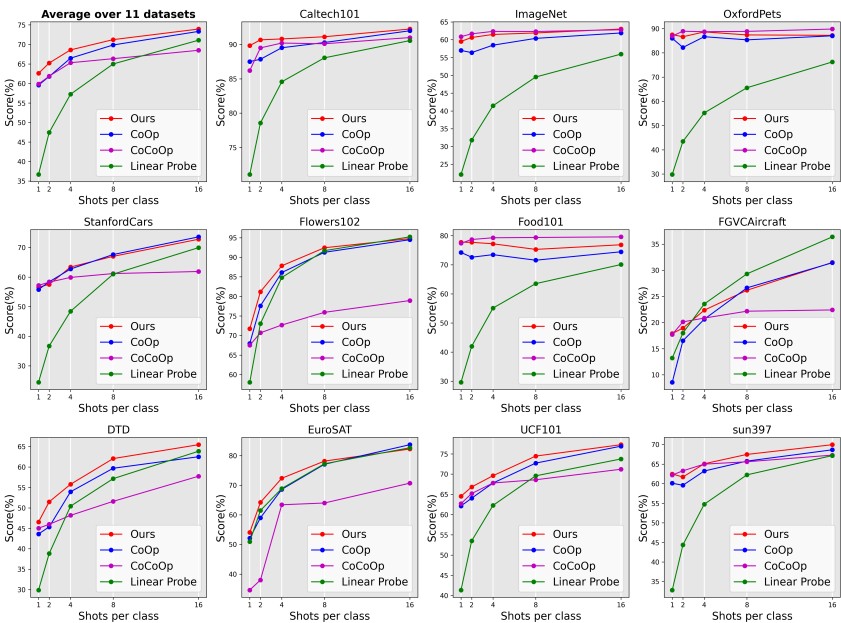

Figure 3: The few-shot learning results on 11 datasets. We compare our PLOT with CoOp, *CoCoOp* , and the Linear Probe method and observe the consistent and significant performance improvement on most datasets. (The average accuracy on all datasets is shown on the left top.)

Table 1: Comparison with CoOp on robustness to domain shift.

| Method | Source | Target | | | |
| --- | --- | --- | --- | --- | --- |
| | ImageNet | -V2 | -Sketch | -A | -R |
| CLIP + CoOp | 61.91 | 54.26 | 32.47 | 21.78 | 54.21 |
| CLIP + PLOT ($N=4$) | **63.01** | **55.11** | **33.00** | **21.86** | **55.61** |

schedule, and a warmup trick with 1e-5 learning rate. Besides, we also followed the epoch strategy to train more epochs for more shots.

We apply $N = 4$ prompts for each category and use $M = 7 \times 7$ due to the feature map size. We set the hyper-parameters in the Sinkhorn distances algorithm [6] as $\lambda = 0.1$ for all the datasets. We set the maximum iteration number of the inner loop as 100 and will early stop the iteration when the average absolute update value $\Lambda < 0.01$. We initialize all values in the vector $v$ and $\mu$ as $1/N$ and $1/M$ respectively. All models are conducted on the Pytorch [34] 1.7.1 and trained on 4 NVIDIA A100 GPUs. We repeated the experiments three times with different seeds and reported the average.

## 4.3 Comparison With CoOp

In this subsection, we compare our PLOT with the baseline CoOp on the few-shot recognition and domain generalization tasks.

**Few-Shot Learning** We summarized the experimental results in Figure 3 where the red line denotes our PLOT method, the blue one denotes CoOp, *the purple line denotes CoCoOp,* and the green one is the CLIP linear probe. The detailed accuracy can be found in the supplementary materials. We observed that both prompt learning methods (PLOT and CoOp) outperform the linear probe method by a large margin. Besides, PLOT can further improve the performance of CoOp and *CoCoOp* on most of the datasets. Taking the average accuracy (at the left top) as the example, Plot respectively gained $3.03\%, 3.45\%, 2.13\%, 1.38\%, 0.61\%$ performance boost over CoOp at $1, 2, 4, 8, 16$ shots. We found the performance gap will reduce when shots increase. It is not surprising since both CoOp and PLOT focus on utilizing the pre-trained knowledge, and the effect of pre-training diminishes given more training data. Among all datasets, PLOT achieves a larger improvement over CoOp on the FOOD101 and DTD datasets and achieves comparable performance only on the StanfordCars datasets. For the FGVCAircraft dataset in which the CoOp only obtains $7.77\%$ accuracy, our PLOT can achieve an accuracy of $17.79\%$, twice as high as that of the CoOp. Note that we don't use the class-specific context, thus the performance on the fine-grained classification datasets is lower, e.g.

Table 2: ***Ablation studies on few-shot recognition***. *PLOT is our defined model with $N = 4$, CoOp is the baseline method, M denotes that we respectively match the global visual feature and multiple textual prompts, V denotes that we apply a constraint to add the variance of prompts, M indicates using the visual feature map instead of the global visual feature.*

| Dataset | Settings | 1 shot | 2 shots | 4 shots | 8 shots | 16 shots |
|---|---|---|---|---|---|---|
| | PLOT | **89.83 ± 0.33** | **90.67 ± 0.21** | **90.80 ± 0.20** | **91.54 ± 0.33** | **92.24 ± 0.38** |
| | CoOp | 87.51 ± 1.02 | 87.84 ± 1.10 | 89.52 ± 0.80 | 90.28 ± 0.42 | 91.99 ± 0.31 |
| Caltech101 | G | 88.13 ± 0.36 | 86.98 ± 1.25 | 88.45 ± 0.79 | 90.16 ± 0.22 | 90.72 ± 0.18 |
| | G+V | 88.28 ± 0.43 | 87.72 ± 1.25 | 88.45 ± 0.30 | 89.82 ± 0.20 | 92.00 ± 0.13 |
| | M | 69.78 ± 1.75 | 71.57 ± 1.59 | 77.18 ± 2.16 | 81.77 ± 0.47 | 86.21 ± 0.20 |
| | M+V | 66.11 ± 8.29 | 71.45 ± 3.98 | 79.30 ± 3.96 | 86.96 ± 0.78 | 89.80 ± 0.17 |
| | PLOT | **46.55 ± 2.62** | **51.24 ± 1.95** | **56.03 ± 0.43** | **61.70 ± 0.35** | **65.60 ± 0.82** |
| | CoOp | 43.62 ± 1.96 | 45.35 ± 0.31 | 53.94 ± 1.37 | 59.69 ± 0.13 | 62.51 ± 0.25 |
| DTD | G | 45.12 ± 1.69 | 48.39 ± 2.08 | 54.75 ± 0.48 | 60.15 ± 0.70 | 63.59 ± 0.76 |
| | G+V | 45.90 ± 2.00 | 48.50 ± 0.99 | 53.96 ± 0.48 | 59.69 ± 1.01 | 63.51 ± 0.66 |
| | M | 13.18 ± 4.57 | 12.25 ± 3.86 | 13.00 ± 4.73 | 20.76 ± 5.42 | 26.99 ± 1.98 |
| | M+V | 12.61 ± 5.93 | 15.11 ± 1.81 | 20.35 ± 1.33 | 44.13 ± 2.39 | 56.85 ± 0.54 |
| | PLOT | **77.74 ± 0.47** | **77.70 ± 0.02** | **77.21 ± 0.43** | **75.31 ± 0.30** | **77.09 ± 0.18** |
| | CoOp | 74.25 ± 1.52 | 72.61 ± 1.33 | 73.49 ± 2.03 | 71.58 ± 0.79 | 74.48 ± 0.15 |
| FOOD101 | G | 74.63 ± 0.11 | 70.15 ± 0.49 | 70.41 ± 0.46 | 70.72 ± 0.98 | 73.68 ± 0.46 |
| | G+V | 74.83 ± 0.31 | 70.09 ± 0.85 | 70.86 ± 0.22 | 70.80 ± 0.68 | 73.93 ± 0.35 |
| | M | 52.02 ± 4.86 | 46.12 ± 1.46 | 46.86 ± 1.39 | 53.43 ± 0.88 | 61.28 ± 0.23 |
| | M+V | 46.52 ± 1.15 | 45.95 ± 2.66 | 53.57 ± 0.83 | 62.95 ± 0.37 | 67.63 ± 1.11 |

Table 3: *Parameter analysis for the number of prompts*

| Dataset | Settings | 1 shot | 2 shots | 4 shots | 8 shots | 16 shots |
|---|---|---|---|---|---|---|
| | N=1 | 88.47 ± 1.15 | 89.19 ± 0.39 | 89.70 ± 0.38 | 90.45 ± 0.24 | 91.56 ± 0.14 |
| Caltech101 | N=2 | 88.86 ± 0.51 | 89.60 ± 0.10 | 90.60 ± 0.17 | 91.25 ± 0.65 | 91.89 ± 0.36 |
| | N=4 | **89.83 ± 0.33** | **90.67 ± 0.21** | 90.80 ± 0.20 | **91.54 ± 0.33** | **92.24 ± 0.38** |
| | N=8 | 89.74 ± 0.30 | 90.18 ± 0.46 | **91.02 ± 0.18** | 91.28 ± 0.28 | 92.04 ± 0.29 |
| | N=1 | 43.91 ± 0.65 | 48.21 ± 2.20 | 53.69 ± 1.10 | 58.90 ± 0.19 | 62.85 ± 0.74 |
| DTD | N=2 | 45.59 ± 2.46 | 48.06 ± 1.92 | 55.58 ± 1.71 | 61.56 ± 0.17 | 64.60 ± 0.92 |
| | N=4 | 46.55 ± 2.62 | 51.24 ± 1.95 | **56.03 ± 0.43** | 61.70 ± 0.35 | **65.60 ± 0.82** |
| | N=8 | **46.89 ± 1.94** | **51.87 ± 2.06** | 54.45 ± 0.48 | **62.20 ± 0.56** | 65.25 ± 0.38 |
| | N=1 | 75.96 ± 0.48 | 76.12 ± 0.59 | 77.11 ± 0.41 | 76.56 ± 0.69 | 77.43 ± 0.80 |
| FOOD101 | N=2 | 77.12 ± 0.49 | 76.89 ± 0.23 | 76.16 ± 0.52 | 75.23 ± 0.69 | 76.81 ± 0.50 |
| | N=4 | 77.74 ± 0.47 | 77.70 ± 0.02 | 77.21 ± 0.43 | 75.31 ± 0.30 | 77.09 ± 0.18 |
| | N=8 | **78.05 ± 0.15** | **78.19 ± 0.07** | **78.12 ± 0.17** | **76.63 ± 0.22** | **77.48 ± 0.12** |

the performance of both CoOp and PLOT without class-specific context is lower than the linear probing on FGVCAircraft. All these performance comparisons can serve as experimental evidence to demonstrate that multiple local prompts and optimal transport distance facilitate the prompt learning of vision-language models. *On StanfordCar, learning multiple prompts didn't significantly improve the performance over a single prompt. It may be because the discriminative characters in this dataset coincide with each other, such that one global prompt and one global visual feature can work well.*

**Domain generalization** The robustness also plays a critical role in model applications since the real-world environment may have large domain shifts with the training data. Therefore, we conducted a robustness evaluation to investigate the transferability of models learned by PLOT.

Table 1 summarizes the results of our PLOT method and CoOp on four ImageNet-based robustness evaluation datasets. For both methods, we trained the models on ImageNet with 16 shots per class. For PLOT, we set the number of prompts as $N = 4$. We can observe that PLOT outperforms CoOp consistently on both source and target domains. These experimental results demonstrate that the performance improvement of our learning multiple prompts doesn't rely on single-domain overfitting.

### 4.4 Ablation Studies and More Analysis

In this subsection, we conducted the ablation studies to investigate the effectiveness of different components, in order to answer the following questions.

**Q: Can we directly learn multiple prompts by respectively matching each prompt with the global visual feature? A: No**. As shown in Table 2, we report the performance of directly matching the global visual feature (notated as "G") and compare it with the baseline CoOp and our PLOT on three datasets including Caltech101, DTD, and FOOD101. We observe that there is no improvement over the baseline on some datasets (such as Caltech101 and FOOD101) if we only directly match prompts and global features. Though "G" obtained the improvement on the DTD dataset, this improvement is still less than that of PLOT. It is because this "G" method is incentivized to learn the indistinguishable prompts, which contradicts our purpose to learn multiple comprehensive prompts. We further add some constraints to push away the prompt from each other. For example, we add an objective function to add the distance between every two prompts as a regularization term, which is notated as "V". However, comparing "G" and "G+V", we do not find significant and consistent improvement when using variance loss.

**Q: Does the improvement mainly come from using all feature maps? A: No**. In PLOT, we apply all feature maps of the visual encoder branch, where each feature is a local embedding at one spatial position. Compared with the global feature, these local features are more informative and contain fine-grained clues. However, we demonstrate that the improvement of PLOT does not only rely on using all feature maps. On the contrary, directly using the feature map to replace the global feature causes a large performance drop. For example, on all three datasets, directly using the feature map ("M" or "M+V") has an around $20\%$ 1 shot accuracy drop over using the global visual feature. It is not surprising since the original CLIP model is trained by matching the global visual feature and language feature. Without using the OT method, the distance between the feature map and multiple textual prompts degenerates to the mean distance of each feature-prompt pair. Besides, when using the feature map, adding the variance loss works well, especially for more shots. For example, the accuracy on 16 shots DTD is improved by a large margin (from 26.99 to 56.85).

**Q: How many prompts are needed? A: 4 prompts are enough** One important hyper-parameter in PLOT is the number of prompts. To analyze the effect of the number of prompts, we conducted the experiments on three datasets with $1, 2, 4, 8$ prompts. The results are summarized in the white part of Table 3. We can observe that the performance obviously increases when adding the number of prompts from 1 to 4. For example, PLOT (N=4) respectively obtains $1.36\%$, $2.64\%$, and $1.68\%$ 1-shot accuracy improvement over PLOT (N=1) on three datasets. Besides, when we further increase the number of prompts, the improvement is not consistent. To balance the improvement and cost, we set $N = 4$ as the default configuration of our PLOT model. In the experiments, we tuned this hyper-parameter on the Caltech101 dataset and applied it to other datasets.

**Q: Can PLOT benefit zero-shot learning? A: No**. CLIP [39] shows that manually designing the prompts can still achieve good performance. We obtain 7 prompts by prompt engineering on the ImageNet dataset and can further ensemble them to obtain **60.38**% top 1 accuracy. In this section, we replace the cosine distance between the global visual feature and prompt ensemble with the OT distance between the feature map and all 7 prompts. However, without any learning, the OT distance only obtains **58.78**% accuracy. *It is a limitation of the PLOT to still need few-shot data for optimization, which cannot be directly applied in the zero-shot setting. We argue there are two reasons why the OT distance does not work without learning: 1) prompt engineering selects prompts based on the global feature and cosine distance, instead of OT distance with feature map; 2) all these selected prompts are closed to the global feature and lack the complementarity.*

**Q: Can PLOT benefit Adapter-based methods? A: Yes**. Adapter-based methods [10, 58] is another research direction of the efficient adaptation of pre-trained vision-language models. Different from the prompt learning that fixes the model parameters and tunes the language prompt, adapter-based methods [10, 58] allow for fine-tuning a part of the network or adding an extra model for training. Recently, adapter-based methods also achieve good performance on few-shot visual recognition. Therefore, we want to explore whether our PLOT method can benefit them, and how.

We apply the Tip-adapter-F [58] as our baseline method, which learns a $Linear(d, N_{cls} \times K_{shots})$ model to describe one image by the similarity with all training samples, where $d$ is the dimension of visual feature, $N_{cls}$ is the number of categories (e.g. 1000 in ImageNet), and $K_{shots}$ is the number of shots. Then, the final similarity consists of the original distance between the visual feature and prompt ensembling and the new distance calculated by the learned feature and one-hot vector of labels (whose dimension is $(N_{cls} \times K_{shots}, N_{cls})$). Please find details in Tip-adapter-F [58]. To introduce PLOT to this framework, we first used the feature map to replace the global feature and

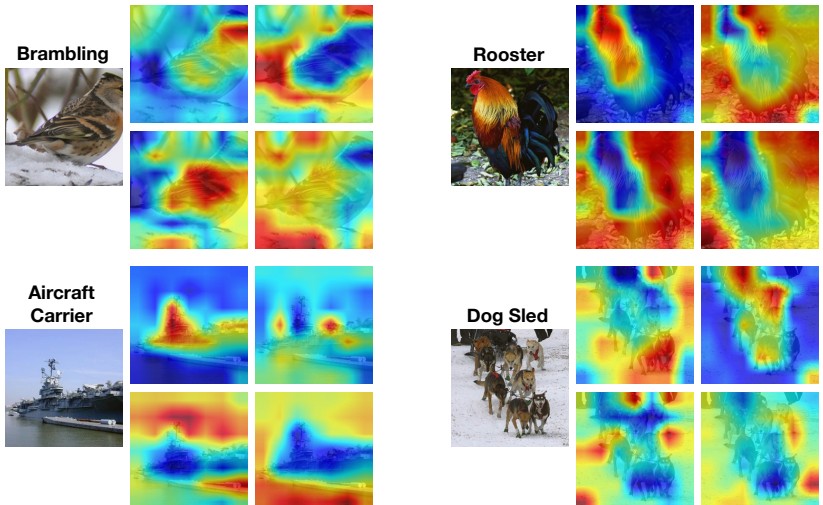

Figure 4: *Visualizations. We provide the heatmaps of transport plan $T$ related to each prompt on 4 categories in ImageNet. Different transport plans focus on different attributes of the object.*

Table 4: Comparison with Adapter-based method.

| Dataset | Methods | 1 shot | 2 shots | 4 shots | 8 shots | 16 shots |
|---------|---------|--------|---------|---------|---------|----------|
| | Tip-Adapter-F | 61.32 | 61.69 | 62.52 | 64.00 | 65.51 |
| ImageNet | Tip-Adapter-F + OT | 61.44 | 61.98 | 62.86 | 64.13 | 65.76 |
| | Tip-Adapter-F + PLOT | **62.27** | **64.31** | **63.89** | **65.04** | **66.17** |

then learned multiple linear models. As a result, with different local features and different linear models, we can obtain a $M \times N$ distance matrix and apply the Sinkhorn algorithm [6] to calculate the OT distance. Furthermore, we can apply the learned prompts as co-partner of the ensembling prompt to refine the final similarity.

Table 4 summarizes the few-shot recognition results of the original Tip-Adapter-F method and our adapter-based PLOT methods on ImageNet. From this table, We observe that using the OT distance can improve the performance of the adapter-based method. Using the learned prompts, we can further promote the accuracy of all settings.

**Q: What is the extra computation time cost of PLOT over CoOp baseline? A: Around** $10\%$ **inference speed and** $5\%$ **training time.** *Despite the performance improvement, the extra computation cost is still a limitation of PLOT. Please see the detailed analysis in the supplementary materials.*

### 4.5 Visualization

In this subsection, we provide some visualization examples of the transport plans $T$ related to different prompts (N=4). We translate each transport plan into colorful heatmaps and resize them into their original size and combine them with the raw image. As shown in Figure 4, we provide the heatmaps of 4 categories in ImageNet. We observe that different transport plans highlight different regions of the image, which demonstrates that the learned multiple prompts are complementary. For the class "Brambling", the prompts respectively focus on the head, tail, wing, and environment. For "Dog Sled", the prompts are related to dogs, the sled, some ties, and the snow environment.

## 5 Conclusion

In this paper, we present a method, named PLOT, to learn multiple comprehensive prompts to describe diverse characteristics of one category. To avoid convergence to one point, we propose to apply the optimal transport to achieve the fine-grained alignment between both vision and language domains. We apply a two-stage optimization strategy where the inner loop fixes the prompts and learns the transport plan to calculate the cross-modality distance, and the outer loop uses this distance to optimize the prompt learner. We build our method on the base of CoOp and achieve significant improvement on the few-shot recognition task in various datasets, which demonstrates the advantage to learn multiple prompts instead of a single one.

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
