# OpenReview forum: "Prompt Learning with Optimal Transport for Vision-Language Models"
_NeurIPS.cc/2022/Conference — NeurIPS 2022 Submitted_

### Official Review · Reviewer_c3r2 · 2022-07-11

**Rating:** 7
**Confidence:** 5
**Soundness:** 4 excellent
**Presentation:** 3 good
**Contribution:** 3 good

**Summary:**

This paper focuses on the few-shot prompt learning for vision-language model CLIP.
On the top of the recent work CoOp who only learns one single prompt, this work proposes a novel method called PLOT, which learn multiple prompts to leverages more fine-grained information from image's feature map. The key motivation of PLOT is to generate a set of prompts so that each prompt can help the model to focus on some specific areas on the feature map. To realize the idea, optimal transport method is used to contrast the effects of each prompt.
The experiment shows that PLOT can achieve better performance over CoOp in most of the datasets.

**Questions:**

None

**Limitations:**

The authors do not include the discussion on the limitation of the method. We hope the authors can provide more analysis on its weakness in discusssion.

**Strengths And Weaknesses:**

Pro:
+ The motivation of PLOT is strong. Figure 1 clearly states the main idea.
+ Using optimal transport to learn distinguishable prompt is reasonable and simple.
+ The experiment shows the effectiveness of the proposed methods.
+ Implementation details are clearly stated.
+ Analysis is comprehensive and strong.


Con:
- Lack of explanation of failure cases, such as StanfordCar in Fig.3. A good illustration and discussion on failure case can help the community know better about your method and the problem.
- Lack of visualization of the learnt prompt. Although it is understandable that it is difficult to print the learnt prompt out into a completed sentence, some words in the prompt can still reveal what the prompt is about. For example, in CoOp, the learnt prompt in OxfordPets has some words like fluffy, paw, etc. Hope PLOT can show more clues about its intrinsic mechanism.

---

> ### Author Response · Authors · 2022-08-02
> **Response to Reviewer c3r2**
>
> Dear Reviewer c3r2,  thanks for your positive comments and helpful suggestions! We made several small revisions, according to your suggestions. Please see the revised paper and supplementary materials for details.
>
> >**Q1:** “ Lack of explanation of failure cases, such as StanfordCar in Fig.3. A good illustration and discussion on failure cases can help the community know better about your method and the problem.”
>
> **A1:** Thanks very much for this helpful suggestion. On the StanfordCar dataset, as you mentioned, learning multiple prompts didn’t significantly improve the performance over a single prompt. It may be because the discriminative characters in this dataset coincide with each other, such that one global prompt and one global visual feature can work well. We have updated this explanation on page 7 of the revised version. To further discover the reason behind this phenomenon, we visualize the attention maps of some failure cases. Specifically, in Figure 1 in the updated version of supplementary materials, we showed two failure examples with class "2000 AM General Hummer" in the StanfordCars dataset. Although the number of prompts is set as 4, we found the learned prompts can be roughly divided into two classes: Foreground and Background, where some of the prompts coincide with each other. For example, in both images,
> prompt 2 (right top) and 3 (left down) focus on the foreground car, while the others focus on the background.  It demonstrates that not all classes have multiple complementary attributes, which motivates us to go further to learn the dynamic local prompts numbers to reduce the computational load in the future. We included the results and discussions in the updated supplementary materials (Section F on pages 4 and 5).
>
> >**Q2:** “Lack of visualization of the learnt prompt. Although it is understandable that it is difficult to print the learnt prompt out into a completed sentence, some words in the prompt can still reveal what the prompt is about. For example, in CoOp, the learnt prompt in OxfordPets has some words like fluffy, paw, etc. Hope PLOT can show more clues about its intrinsic mechanism.”
>
> **A2:** Thanks very much for these suggestions. We produced the visualization results of the learned prompt and the corresponding discussion, but could not put them in the main paper due to the page limits. Please kindly refer to Section E and Table 4 in the supplementary materials.  We found some interesting relations between the learned prompts and corresponding optimal transport plans, which help to understand the semantic attributes of different local prompts. If you think it is essential to show such results in the main paper, we will find a way to achieve it.
>
> >**Q3:** “The authors do not include a discussion on the limitation of the method. We hope the authors can provide more analysis on its weakness in the discussion.”
>
> **A3:** Thank you!  We have explicitly included the limitations in Section 4.4 of the updated version based on previous discussions. There are two main limitations of our method. First, PLOT still needs few-shot data for optimization, which cannot be directly applied in the zero-shot setting. It is because that CLIP is trained by matching the global visual feature and text, while our method applies the local visual features which need few-shot data for optimization. Second, The method needs more computing costs than CoOp.

---

### Official Review · Reviewer_bMB5 · 2022-07-11

**Rating:** 6
**Confidence:** 3
**Soundness:** 3 good
**Presentation:** 3 good
**Contribution:** 3 good

**Summary:**

The authors propose a new prompt learning method, which learns multiple prompts to describe diverse characteristics of categories. To optimize the model effectively, they apply optimal transport to match vision and language modalities. The experiments demonstrate its effectiveness and achieve improvement on few-shot learning tasks.

**Questions:**

Questions
After optimization, for different images and classes, are they sharing the same prompts? If so, there might be issues. Because different images have different local patterns, like 'a dog lying on the ground'  v.s. 'a bird in the forest'. They have totally different local patterns (positions).

Typos:
	1. Line 55, prompts learning -> prompt learning
	2. Line 75, simultaneously several items -> several items simultaneously
	3. Line 90, filed -> field
	4. Line 110, 111: improve -> improves, achieve -> achieves
	5. Line 135, vectorss -> vectors
	6. Line 158, is same as -> is the same as
	7. Line 232, summarize -> summarizes
Line 274, we obtain -> We obtain

**Strengths And Weaknesses:**

Strengths:
	1. The motivation is innovative. Conventional methods use a single prompt which is not able to capture multi-grained features. The authors therefore aim to learn multiple comprehensive prompts to describe different categories.
	2. An interesting combination between optimal transport theory and prompt learning.
	3. The experiments are comprehensive and demonstrate the effectiveness of introduced method.


Weaknesses:
	1. The proposed method is a two-stage optimization strategy, which is a bit difficult to balance the two steps optimization. Could it be end-to-end training?
	2. Although it is intuitive that including multiple local prompts helps, for different categories, the features and their positions are not the same.

---

> ### Author Response · Authors · 2022-08-02
> **Response to Reviewer bMB5**
>
> Dear Reviewer bMB5,  we are sincerely grateful for your careful reading and valuable feedback, which has helped improve our paper. We provide the point-to-point response below and have updated the paper and appendix accordingly.
>
> >**Q1**: "The proposed method is a two-stage optimization strategy, which is a bit difficult to balance the two steps of optimization. Could it be end-to-end training?"
>
> **A1**: Thanks very much for this valuable question. There are two points to this question. First, thanks to the Sinkhorn algorithm[1], the optimizations of optimal transport and prompt learning can be integrated. Though it is a two-stage optimization strategy, we find it is natural and relatively easy to the optimization. We have added this into page 5 of the revised paper to reflect this point. Thanks for your question.
>
> Second, we also updated the paper to make it more explicit that though the optimization strategy is two-stage, the whole training flow is end-to-end. It is because the transport plan is computed using a small number of matrix multiplications using the iterative Sinkhorn algorithm as one feedforward module of the neural network. The gradients of these matrix multiplications are taped for backpropagation for end-to-end optimization, which makes the whole system fully differentiable (including the iterative algorithm) and easy to implement using an autograd library like PyTorch.
>
> Besides, the optimization of the transport plan with the Sinkhorn algorithm is very efficient. As shown in lines 303-305 of the original paper and Section D in the supplementary material, we demonstrate that this optimization process only costs around 10% inference speed and 5% training time.
>
> [1]Cuturi M. Sinkhorn distances: Lightspeed computation of optimal transport[J]. Advances in neural information processing systems, 2013, 26.
>
> >**Q2**: "After optimization, for different images and classes, are they sharing the same prompts? If so, there might be issues. Because different images have different local patterns, like a dog lying on the ground' v.s. 'a bird in the forest'. They have different local patterns (positions)."
>
> **A2**: Thanks for this valuable question.  First, the **prompts** are shared for different images and classes, and the **prompt features** are only shared for different images of the same class (prompt features contain the information of the class label). It requires the distance between prompt features and visual features to be intra-category robust even if the images in the same class are misaligned. In this paper, we learn a transport plan T to align prompt features and local visual features. Let us explain why the optimal transport helps solve this intra-category misalignment problem below.
>
> Suppose we have two prompts “head” and “tail” for the class “bird”, given two images with position shifts where **Image A** is “left head and right tail” and **Image B** is “right head and left tail”. It will provide a wrong similarity score if we directly match the prompt “head” with the left region and prompt “tail” with the right region.  To deal with this problem, we leverage optimal transport, which optimizes a transport plan to adaptively match the prompt feature and visual feature. The optimal transport can acquire the optimal matching flows with minimal distance. In this example, the optimal transport will match prompt “head” with the left region for **Image A** and match prompt “head” with the right region for **Image B**. Thus, even though the prompt features are shared for all images in the same class, the optimal transport can provide the adaptive matching plan to solve the misalignment problem.  We have highlighted this capability of optimal transport on page 2 of the revised paper.
>
> >**Q3**: "Typos."
>
> **A3**: Thank you very much! We have corrected the typos in the revised paper.

---

### Official Review · Reviewer_fGe6 · 2022-07-11

**Rating:** 7
**Confidence:** 5
**Soundness:** 3 good
**Presentation:** 4 excellent
**Contribution:** 3 good

**Summary:**

This paper focused on adapting large-scaled pre-trained vision-language model (i.e., CLIP) to downstream datasets under the few-shot setting. Compared to the recent related work CoOp, this work (1) learns multiple prompts, (2) uses both local features and global features, and (3) measures the similarity of prompts and visual features using optimal transport. Experiments on 11 benchmark visual recognition datasets show that the proposed method outperforms the baseline CoOp. Ablation studies further more analysis on the contribution of each component.

----------------------

Updated after rebuttal:

Reasons to accept:

+ The motivation for this work is clear. Single prompts may lose the details of local attributes or contexts.

+ The idea of applying optimal transport to prompt learning makes sense and meets well with the motivation.

+ The experiments and analysis are comprehensive. The improvement is significant.

I am satisfied with the authors' responses to my comments and concerns, and I increased my rating from 6 to 7. I have read other reviewers' concerns on (1) two-stage training, (2) sharing of prompts, and (3) failure analysis and visualization. I think these questions are not severe for rejection and the authors have responded to these concerns.

**Questions:**

Please see weaknesses in the above.

**Limitations:**

The authors have adequately addressed the limitations.

**Strengths And Weaknesses:**

Overall, considering both strengths (simple but effective pipeline, comprehensive experimental analysis) and weakness (insight and contribution), my initial rating is borderline.

Strengths:

\+ The problem of adapting CLIP under few-shot setting is recent. Compared to the baseline method CoOp, the improvement of the proposed method is significant.

\+ The ablation studies and analysis in Section 4.4 is well organized and clearly written. It is easy to follow the analysis and figure our the contribution of each component. Also, Figure 2 is well designed and clear to illustrate the pipeline.

\+ The experimental analysis is comprehensive. The analysis on computation time and inference speed is also provided.

Weakness:

\- (major concern) The contribution is somehow limited. The main contribution is applying optimal transport for few-shot adaptation of CLIP. After reading the paper, it is not clear enough to me why Optimal Transport is better than other distance. Especially, the insight behind the application of Optimal Transport is not clear. I would like to see more analysis and explanation on why Optimal Transport works well. Otherwise, it seems that this work is just an application work on a specific model and a specific task, which limits the contribution.

\- The recent related work CoCoOp [1] is not compared in the experiments. Although it is a CVPR'22 work that is officially published after the NeurIPS deadline, as the extended version of CoOp, it is necessary to compare with CoCoOp in the experiments.

\- In the approach method, there lacks a separate part or subsection to introduce the inference strategy, i.e., how to use the multiple prompts in the test stage.

\- Table 2 mixed different ablation studies (number of prompts, visual feature map, constraint). It would be great if the table can be split into several tables according to the analyzed component.

\- The visualization in Figure 4 is not clear. It is not easy to see the attention as it is transparent.

References

[1] Kaiyang Zhou, Jingkang Yang, Chen Change Loy, and Ziwei Liu. Conditional prompt learning for vision-language models. In CVPR, 2022.


------------

After reading the authors' response and the revised version, my concerns (especially the contribution of introducing the optimal transport distance for fine-tuning vision-language models) are well addressed and I am happy to increase my rating.

---

> ### Author Response · Authors · 2022-08-02
> **Response to Reviewer fGe6, Part I**
>
> Dear Reviewer fGe6,  we appreciate your time dedicated to reviewing this paper and the valuable comments, which have helped improve our work. We revised our manuscript accordingly (please see the updated paper and supplementary materials for details), and provide a point-to-point response below.
>
> >**Q1**: "it is not clear enough to me why Optimal Transport is better than other distances. Especially, the insight behind the application of Optimal Transport is not clear. I would like to see more analysis and explanation on why Optimal Transport works well."
>
> **A1**: Thanks for your valuable question! In light of your comments, we added more comparisons between Optimal Transport and other distance to show the insight of using Optimal Transport in the revised manuscript.
>
> Conventional prompt learning methods (e.g. CoOp) adopt the **Euclidean distance** to measure the similarity between **single (global) prompt feature vector** and **single (global) visual feature vector**. In contrast, PLOT proposes to use the **Optimal Transport** to calculate the distance between **multiple (local) prompt features** and **multiple (local) visual features**.  Before we discuss the differences between Optimal Transport and conventional Euclidean distance, we would mention the differences between PLOT and conventional prompt learning methods in three folds: multiple (local) prompt features v.s. single (global) prompt feature; multiple (local) visual features v.s. single (global) visual feature; and using Optimal Transport to calculate set-to-set distance v.s. Euclidean distance.
> Let’s discuss them one by one:
>
> **1, Multiple (local) prompt features v.s. single (global) prompt feature:**  Prompt is used to represent the classes. One class can be described with many intrinsic characteristics and extrinsic context relations. It motivates us to use multiple prompt candidates which focus on different attributes to replace one global prompt.  Compared with a single prompt, multiple local ones can comprehensively represent the class and thus facilitate classification.
>
> **2, Multiple (local) visual features v.s. single (global) visual feature:** Compared with the single global visual feature, the information provided by local features are more discriminative and transferable. These local features are more robust to visual misalignment such as within-class position shift.
>
> **3, Using Optimal Transport to calculate set-to-set distance v.s. Euclidean distance:** As shown in lines 44-48 in the original manuscript, if we use multiple prompt features and single visual feature, all prompts are encouraged to be closer to one single point and thus tend to learn the same characteristics. It calls for multiple prompt features and multiple local visual features. Given the motivation that we use multiple local prompt and visual features to replace the global one, we further need to calculate the distance between the two feature sets.  The conventional solution is to calculate the pairwise Euclidean distance and take the mean of all pairs. However, this distance is not robust for the feature shift. In this paper, we propose to apply the optimal transport to align the visual features and prompts, whose insight is to adaptively align different visual features for each local prompt, which is more robust to the visual misalignment and tolerates well feature shift[1]. It is because Optimal Transport distance can represent the data structure by learning an adaptive transport plan for the fine-grained alignment. Besides, some theoretical analyses [2] also show that Optimal Transport distance space is more flexible than Euclidean spaces. Here we show a toy example to show the advantage of the Optimal Transport distance. Given two prompts whose values are 2 and 0, and two visual feature sets of the same class with feature shift (1,1,1,-1) and (1,1,-1,-1).  In the mean Euclidean distance space, we get the distances of 1.5 and 1.25 respectively, while with optimal transport, we get the same distance of 1.
>
> [1]Rubner Y, Tomasi C, Guibas L J. The earth mover's distance as a metric for image retrieval[J]. International journal of computer vision, 2000, 40(2): 99-121.
>
> [2]Frogner C, Mirzazadeh F, Solomon J. Learning Embeddings into Entropic Wasserstein Spaces[C]//International Conference on Learning Representations. 2019.

---

> > ### Comment · Reviewer_fGe6 · 2022-08-06
> > **Author Rebuttal Acknowledgement**
> >
> > Thank the authors for the comprehensive response and updated paper! The detailed response addressed my concern, and I am happy to raise my rating.

---

> > > ### Author Response · Authors · 2022-08-06
> > > **Thanks for raising the rating score**
> > >
> > > Thanks a lot for checking our response and the updated paper. We are glad that our response and updated presentation have resolved your concerns regarding the motivation, contribution, and clarity of the paper. Your valuable comments have improved our presentation a lot and made the paper more readable. Thank you very much!

---

> ### Author Response · Authors · 2022-08-02
> **Response to Reviewer fGe6, Part II**
>
> >**Q2**: "It seems that this work is just an application work on a specific model and a specific task, which limits the contribution."
>
> **A2**: Recently, large-scale pre-trained models have achieved great success and attracted a lot of attention.  As a parallel paradigm with “pretraining-finetuning”, prompt learning shows the great potential to efficiently adopt pre-trained knowledge into downstream tasks. In this paper, we propose learning multiple comprehensive prompts for a more representative prompt learning.  It is a plug-and-play module that can be easily extended to any baseline prompt learning method such as CoOp[3] or Tip-adapter[5]. It can achieve orthogonal and complementary improvement over the baseline methods as shown in Table 2 and Table 3.
>
> Most existing methods[3,4,5] exploring how to adapt the pre-trained vision-language knowledge apply the few-shot recognization as the evaluation metric. It is because that few-shot recognization can effectively evaluate the adaptation ability of the method when only few-shot downstream data is available. Besides, the same few-shot recognization setting can provide a fair comparison with existing methods, such as CoOp[3]. Some recent progress shows that the pre-trained vision-language knowledge can also be applied to other tasks, e.g. DenseCLIP[6] applies the pre-trained model to visual detection and segmentation. Our method also has potential for these applications.
>
> [3]Zhou K, Yang J, Loy C C, et al. Learning to prompt for vision-language models[J]. arXiv preprint arXiv:2109.01134, 2021.
>
> [4]Gao P, Geng S, Zhang R, et al. Clip-adapter: Better vision-language models with feature adapters[J]. arXiv preprint arXiv:2110.04544, 2021.
>
> [5]Zhang R, Fang R, Gao P, et al. Tip-adapter: Training-free clip-adapter for better vision-language modeling[J]. arXiv preprint arXiv:2111.03930, 2021.
>
> [6]Rao Y, Zhao W, Chen G, et al. Denseclip: Language-guided dense prediction with context-aware prompting[C]//Proceedings of the IEEE/CVF Conference on Computer Vision and Pattern Recognition. 2022: 18082-18091.
>
> >**Q3**: "The recent related work CoCoOp is not compared in the experiments. "
>
> **A3**: Thanks very much for this suggestion! We have added the performance comparison in the experiments of the revised paper.  Please see updated Figure 3 for the details. In the original submission, we didn’t add CoCoOp[7] in the experiments because the experimental settings of CoCoOp and our method were different:  CoOp[3] and our method focus on the few-shot recognition task, while CoCoOp focuses on the generalization for the unseen new classes. For a fair comparison, we re-run CoCoOp in our experimental settings with the same backbone network.  The performance on all 11 datasets is shown in the updated Figure 3. We found that in most datasets, our method achieved better performance than CoCoOp. Besides, since CoCoOp focuses on "from base to new" setting, its performance is even lightly slower than CoOp.
>
> [7]Zhou K, Yang J, Loy C C, et al. Conditional prompt learning for vision-language models[C]//Proceedings of the IEEE/CVF Conference on Computer Vision and Pattern Recognition. 2022: 16816-16825.
>
> >**Q4**: "In the approach method, there lacks a separate part or subsection to introduce the inference strategy."
>
> **A4**: We appreciate this suggestion to help improve our paper. We have added a subsection of inference strategy on page 5 of the updated version. Below we give a more detailed introduction of the inference strategy.
>
> During inference, the goal of the method is to classify the query images given the learned prompt and the pre-trained model. Given one image, we first feed it into the image encoder to obtain a set of local visual features.  Then, we combine the learned prompts and the names of classes to get a set of prompt features. Specifically, the visual feature set contains M=HxW, which is 7x7, feature vectors, while the prompt feature set contains NxC feature vectors, where N is the number of local prompts and C is the number of classes (C=1000 in Imagenet). Then, we calculate the distance between the visual feature set and the prompt feature set of each class. To calculate the distance, we optimize the transport T to reweight the distance matrix. This transport T helps align the local visual features and multiple prompts for a fine-grained cross-modal matching. After obtaining the OT distance for each class, we sort the distance and classify the image into the class with minimal OT distance.
>
> >**Q5**: "It would be great if Table 2 can be split into several tables according to the analyzed component." & "The visualization in Figure 4 is not clear."
>
> **A5**: Thank you! This suggestion helped a lot to improve the readability of the paper. We split Table 2 into 2 tables for ablation studies and parameters analysis respectively. Then we darkened the color of the attention map in Figure 4 for clarity. Please kindly see pages 7 and 9 in the revised paper for details.

---

### Author Response · Authors · 2022-08-02
**Response to all reviewers**

We thank all reviewers for their thoughtful and constructive review of our manuscript. We were encouraged to hear the reviewers think PLOT is “effective”(Reviewer fGe6), “reasonable”(Reviewer c3r2), and “well-motivated” (Reviewers bMB5 and c3r2), and that the effectiveness can be demonstrated with “comprehensive experimental analysis”(Reviewers fGe6, bMB5, and c3r2). We provide a general response here to summarize the modifications of the manuscript.

- To reviewer fGe6: We have clarified the motivation and advantage of using Optimal Transport on page 2.
- To reviewer fGe6: We have added the experimental comparison with CoCoOp in Figure 3.
- To reviewer fGe6: We have added a subsection on page 5 to introduce the inference strategy of PLOT.
- To reviewer fGe6: We have split Table 2 into 2 tables for ablation studies and parameters analysis respectively.
- To reviewer fGe6: We have refined Figure 4 for clarity.
- To reviewer bMB5: We have clarified our optimization strategy on page 5.
- To reviewer bMB5: We have highlighted the capability of optimal transport for misalignment on page 2.
- To reviewer bMB5: We have corrected the typos accordingly.
- To reviewer c3r2: We have updated the explanation for the failure cases on page 7 and included more visualization results of the failure cases in Section F of the supplementary materials.
- To reviewer c3r2: We have explicitly included the limitations in Section 4.4 based on previous discussions.

---

### Meta-Review · Area_Chair_eGXx · 2022-08-24

**Recommendation:** Reject
**Confidence:** Certain

**Metareview:**

This paper presents a novel perspective of prompt tuning for few-shot visual recognition: a dynamic matching algorithm between the prompt candidate and the visual features. Compared to the existing CoOp and CoCoOp algorithm, the proposed "Optimal Transportation" idea definitely sounds better and indeed achieves better performance. All the reviewers acknowledge the merits of the paper.

Though AC also acknowledges the merits of the paper, there are two unaddressed demerits:

1) As the proposed method is essentially an ensemble method, comparisons with prompt ensemble should be conducted. Unfortunately, the reported "G" in Table 2 and Line 254-265 is not a proper ensemble, because the "G" baseline may degenerate into many duplicate single CoOp models with different random seeds. AC conjectures that this is the reason why G's performance is only slightly different from CoOp. By "proper ensemble", the authors may want to try initializations by "this is a photo"+"a picture of" + "there is an xx of" etc, or, augmenting each image by random crops, each of which corresponds to a "G", and then ensemble.

2) This paper lacks an important "Base to New" setting as proposed by CoCoOp. As the proposed PLOT in this paper has significantly more tunable parameters, AC doubts that it may lead to the overfitting of the training classes but ruins (or forgets) other classes which were used to have good zero-shot performance without training.

Unfortunately, AC regrets to recommend reject and wishes the best of luck in re-submitting the paper to other venues.

**Award:**

No

---

### Decision · Program_Chairs · 2022-09-14

Reject